

**Insights of warm cloud biases in CAM5 and CAM6 from the single-column modeling**
**framework and ACE-ENA observations**
**Yuan Wang[1,*], Xiaojian Zheng[2], Xiquan Dong[2], Baike Xi[2],  and Yuk L. Yung[3]**
[1]Department of Earth, Atmosphere, and Planetary Sciences, Purdue University, West Lafayette,
IN, USA
[2]Department of Hydrology and Atmospheric Sciences, University of Arizona, Tucson, AZ, USA
[3]Division of Geological and Planetary Sciences, California Institute of Technology, Pasadena,
CA, USA
*Corresponding author: Yuan Wang (yuanwang@purdue.edu)





**Abstract**

There has been a growing concern that most climate models predict too frequent precipitation, likely due to lack of reliable sub-grid variability and vertical variations of microphysical processes in low-level warm clouds. In this study, the warm cloud physics parameterizations in the singe-column configurations of NCAR Community Atmospheric Model version 6 and 5 (SCAM6 and SCAM5, respectively) are evaluated using ground-based and airborne observations from the DOE ARM Aerosol and Cloud Experiments in the Eastern North Atlantic (ACE-ENA) field campaign near the Azores islands during 2017-2018. Eight-month SCM simulations show that both SCAM6 and SCAM5 can generally reproduce marine boundary-layer cloud structure, major macrophysical properties, and their transition. The improvement of warm cloud properties from CAM5 to CAM6 physics can be found compared to the observations. Meanwhile, both physical schemes underestimate cloud liquid water content, cloud droplet size, and rain liquid water content, but overestimate surface rainfall. Modeled cloud condensation nuclei (CCN) concentrations are comparable with aircraft observed ones in the summer but overestimated by a factor of two in winter, largely due to the biases in the long-range transport of anthropogenic aerosols like sulfate. We also test the newly recalibrated autoconversion and accretion parameterizations that account for vertical variations of droplet size. Compared to the observations, more significant improvement is found in SCAM5 than in SCAM6. This result is likely explained by the introduction of sub-grid variations of cloud properties in CAM6 cloud microphysics, which further suppresses the scheme sensitivity to individual warm rain microphysical parameters. The predicted cloud susceptibilities to CCN perturbations in CAM6 are within a reasonable range, indicating significant progress since CAM5 which produces too strong aerosol indirect effect. The present study emphasizes the importance of understanding biases in cloud physics parameterizations by combining SCM with in situ observations.



**1. Motivation and Background**

Marine boundary-layer (MBL) clouds are crucial for the global radiation budget, as they efficiently regulate the solar radiation reaching the ocean surface and largely determine the climate sensitivity (Dong et al., 2022; Sherwood et al., 2020). However, numerical simulations of MBL clouds in global climate models (GCM) remain challenging, mainly due to the mismatch of the spatial scales of MBL clouds (tens of meters) and GCM grids (~ 100 km). Therefore, empirical parameterizations of subscale cloud properties and variabilities, for both microphysics and macrophysics, play a critical role in predicting MBL clouds and precipitation in GCM (Wang et al., 2013). Consequently, how to constrain and improve those cloud parameterizations using the state-of-the-art observations become an important issue. One challenging aspect of the GCM cloud evaluation lies in the tight coupling between cloud physics and dynamics, as cloud microphysics can feedback to dynamics and thermodynamics through heating profile alteration or radiation flux interference (Wang et al., 2014, 2020).

To better probe the uncertainty source in the cloud physical parameterizations, a simplified GCM configuration has been developed to separate cloud physics from large-scale dynamical and thermodynamical conditions. The so-called single column model (SCM) is ideal for utilizing in situ observations from the field campaigns that are normally conducted intensively over the targeted area (Zhao et al., 2021). The modeling framework adopted in this study, NCAR Community Earth System Model (CESM), has a long history of providing such a modeling tool along with the development of its comprehensive models (Liu et al., 2007; Gettleman et al., 2019). With more added features and enhanced representations of cloud and aerosol in the cloud physical parametrizations in CESM version 1 and 2, it is valuable to evaluate the single-column versions of them using the recent field measurements.

The Eastern North Atlantic (ENA) is an ideal place around the world to study MBL clouds, considering the prevailing MBL cloud occurrence, diverse mesoscale meteorological conditions (Jensen et al., 2021; Zheng et al., 2022a), and distinctive aerosol sources (Wang, J. et al., 2021). A recent field campaign, the Aerosol and Cloud Experiments in the Eastern North Atlantic (ACE-ENA) provide ample ground-based and in situ aircraft observations of cloud micro- and macrophysics, aerosol properties, as well as atmospheric states over a whole summer and winter (Wang, J. et al. 2021; Wu et al., 2020). Recent WRF large-eddy simulations (LES) driven by the ERA5 reanalysis over the ENA well reproduce the general vertical variations of meteorological



factors and cloud cellular structure (Wang et al., 2020). Meanwhile, LES and observations exhibit
substantial discrepancies in the evolution of MBL clouds in two selected stratocumulus cases
during the ACE-ENA field campaign, likely due to the biases in both warm cloud physical
parameterizations and large-scale forcing. Those issues motivate us to look for stronger
observational constraints in the single-column framework which minimizes the propagated errors
from large-scale forcing. In this study, we use the ARM 3-hourly large-scale forcing of
atmospheric states specifically developed for the ACE-ENA Intensive Observation Periods (IOP)
to drive SCM.

The uncertainties of warm cloud physics in the atmospheric component of CESM1/2 have

been reported in many previous studies (e.g. Kay et al., 2016; Zhao et al., 2022), while most of
them focused on addressing the issues on the global scale. Leveraging the continuous radar
retrievals of MBL cloud and drizzle microphysical properties during ACE ENA, Dong et al. (2021)
modified the parameterizations of two key processes in warm cloud microphysics in CAM5, i.e.,
autoconversion from cloud droplets to rain drops and accretion of cloud droplets by raindrops.
They showed that by applying this set of new parameterizations to CAM5 in global climate
simulations, precipitation frequency is generally reduced but with enhanced intensity mainly in the
mid-latitude regions, alleviating the long-lasting issue in the climate models, e.g., "too frequent
and too light precipitation". Even the cloud radiative effect and top-of-atmosphere radiative flux
simulations can be improved consequently. Therefore, a remaining and outstanding question lies
in whether such a new scheme works well over the location where the radar observational
constraints come from originally. The single-column modeling framework enables us to examine
the effect of the modified microphysical scheme on the local scale.
**2. Methodology**
**2.1 Single column version of Community Atmospheric Model**

In this study, we use single-column configuration of Community Atmospheric Model

version 6 (referred to as SCAM6 thereafter) in the Community Earth System Model (CEMS 2.1.1).
NCAR CESM is a community GCM that has been widely used to study the climate change (Yeager
et al., 2018), precipitation extremes (Wang et al., 2016), cloud processes (Wang et al., 2018), and
aerosol-cloud-radiation-circulation feedbacks in the Earth system (Wang et al., 2015). The
atmosphere component of CESM2 (CAM6) has been modified substantially with a range of
enhancements and improvements for the representation of physical processes since its last version,



CAM5. In particular, the modifications on the aerosol and cloud parameterizations are extensive.
For example, a multivariate PDF-based third-order turbulence closure parameterization scheme,
Cloud Layers Unified By Binormals (CLUBB), is implemented to unify the representation of
boundary layer, shallow convection, and stratiform macrophysics in the model (Bogenschutz et al.,
2013; Golaz and Larson, 2002). The two-moment cloud microphysical scheme is updated to its
version 2 (Gettelman and Morrison, 2015) by incorporating prognostic precipitation (rain and
snow), sub-stepping technique, and re-tuned autoconversion scheme which is critical for aerosol
indirect effect on cloud lifetime and precipitation (Malavelle et al., 2017). The strong coupling
between CLUBB and MG2 also facilitates cloud-aerosol-environment interactions. Deep
convection remains parameterized by the Zhang-McFarlane (1995) scheme and has been re-tuned
to increase the sensitivity to convective inhibition, which could potentially signify the impact of
absorbing aerosols within the planetary boundary layer (PBL). Parameterizations of homogeneous
ice nucleation and heterogeneous immersion nucleation in cirrus clouds (Liu and Penner, 2005)
explicitly consider the effects of sulfate and dust aerosol serving as ice nuclei on the cold clouds.

The aerosol module in CESM is updated from a three-mode to four-mode approach

(MAM4) to better consider the aging processes of black carbon in the atmosphere (Liu et al., 2016;
Wang et al., 2018). Six types of aerosols with different hygroscopicity and optical properties are
considered in MAM3, including sulfate, black carbon (BC), primary organic matter (POM),
secondary organic aerosol (SOA), dust and sea salt. The aerosol module accounts for most of the
important processes associated with atmospheric aerosols, including emission, nucleation,
coagulation, condensational growth, gas and aqueous-phase chemistry, dry deposition, in-cloud
and below-cloud scavenging, re-production from evaporated cloud droplets and suppression, as
well as agricultural, deforestation, and peat fires (Li and Lawrence, 2017). To test the impacts of
cloud physical parameterization on the model fidelity, we also conduct the single-column
simulations using the CAM5 physics (SCAM5) under the same large-scale forcing data.

Because the ACE-ENA is a relatively new field campaign and does not have a pre-defined

case in SCAM6, we create a new case in CAM6 based on a new set of large-scale data for this
IOP. To cover the full IOP in our simulations, we run SCAM over 8 months from June 1, 2017, to
Feb 1, 2018. The large-scale forcing over the ARM-ENA is developed from the constrained
variational analysis (VARANAL, Xie et al., 2004; Tang et al., 2019). VARANAL is based on
ERA5 reanalysis (Copernicus Climate Change Service, 2017) with the additional input of





observations from the ARM ENA site incorporated into the variational analysis, to represent the
atmospheric states over a Global Climate Model (GCM) grid box. The original VARANAL data
is produced specifically for the ACE-ENA IOP, with a temporal resolution of 3-hour and 45
vertical levels.

To minimize the biases in aerosol advection and dynamical forcing, aerosol and the

temperature fields are nudged to their initial conditions on different timescales, varying from 10
days at the bottom of the model to 2 days at the top of the model (Gettelman et al., 2019). Also, to
simulate the right seasonal variations of aerosol and temperature initial conditions, each of our
model integration only lasts one month, and a new sequential run will follow with updated initial
conditions. By doing so the seasonality of aerosols will follow that of climatology on the monthly
basis.
**2.2 ACE-ENA observations**

Aircraft in situ observations during the ACE-ENA provide best available characterizations

of cloud and aerosol vertical distributions, with differentiation of aerosol types and hygroscopicity.
During the two IOPs, 39 flights were deployed to collect data for 39 days, 20 in the summer IOP,
19 in the winter IOP. Meanwhile, ground-based observations were conducted simultaneously and
consecutively. Based on the Ka-band ARM Zenith Radar (KAZR) measurements, cloud and rain
microphysical properties (cloud droplet effective radius, $r_c$; cloud droplet number concentration,
$N_c$; cloud liquid water content, $CLWC$; rain droplet mass median radius, $r_{m,r}$; rain droplet number
concentration, $N_r$; and rain liquid water content, $RLWC$) over the ARM ENA site can be retrieved
(Wu et al. 2020). The cloud and drizzle microphysical retrievals were validated by the aircraft in-
situ measurements from ACE-ENA field campaign, with the estimated median uncertainties of
~15% for $r_c$; ~30% for $r_{m,r}$; and ~50% for $N_r$ and $RLWC$. Note that the subscript "c" denotes
cloud and subscript "r" denotes rain. The model counterparts are extracted and compared with the
retrieval, except the $r_{m,r}$ which is not an output from the model. Following the method in Wu et al.
(2020) equation 2a, the $r_{m,r}$ can be calculated by:
$$r_{m,r} = \left(\frac{RLWC*3.67^4}{\rho_w*N_W*8\pi}\right)^{1/4} \qquad (1),$$
where the $\rho_w$ is water density, and the $N_W$ is the normalized drizzle number concentration ($N_W =$
$N_r/r_{m,r}$). Furthermore, the $CLWC$ and $RLWC$ are scaled by the cloud (rain) fraction within the
grid box to match the retrievals.



For the aircraft in-situ measurements, the Passive Cavity Aerosol Spectrometer (PCASP)
measured the aerosols with the size range from 0.1 µm to 3.2 µm (Goldberger, 2020), hence the
accumulation mode aerosol number concentration ($N_{Acc}$) can be derived from the PCASP 0.1 µm
to 1.0 µm measurement. The CCN number concentration ($N_{CCN}$) is obtained by the CCN-200
particle counter on board the G-1 aircraft. The $N_{CCN}$ is a measurement under the controlled
supersaturation of 0.35 % with a humidified particle size range from 0.75 to 10 µm (Uin and Mei,
2019). The PM1 aerosol chemical components mass concentrations are measured by the Aerodyne
High-Resolution Time-of-Flight Aerosol Mass Spectrometer (HR-ToF-AMS). The accuracy of
each individual instrument can be found in the instrument handbooks available on the ARM
website. To make better comparisons, this study only selects the research flights with a 'L' shape
pattern center at the ARM-ENA site. The SCAM6 samples are selected within each time duration
of the aircraft cases. Note that the aircraft cases are selected up to end of Jan 2018 due to the end
of SCAM6 simulations. To ensure the apple-to-apple comparison between model and observations,
the cloud and rain samples are selected following the same criteria: 1) $4\ \mu m < r_c < 25\ \mu m$; 2)
$CLWC > 0.01\ gm^{-3}$; 3) $N_c > 1\ cm^{-3}$; and 4) $RLWC > 1 \times 10^{-4}\ gm^{-3}$. The geopotential
height from the model output is extracted for each time step, hence the quantities at pressure level
can be converted to height level and compared with the observation results. Both model and
observation results are limited to below 3km.
**3. Evaluation of SCAM6 using ACE-ENA observations**
**3.1 Meteorological conditions**
To understand the cloud and drizzle property differences between simulations and
observations, we first evaluate the SCAM6 simulated meteorological conditions by the ARM
Interpolated Sonde (INTERPSONDE) value-added product (VAP), which is an independent
dataset from the large-scale forcing data used to drive the SCM. As shown in Fig. 1, the simulated
air temperature ($T_{air}$) values are comparable to the observed ones with clear seasonal variations.
The statistics from the 8-month simulations shows that the differences in both mean and median
$T_{air}$ agree within 1% to the observed ones, supporting the high fidelity of the model to reproduce
the temperature field. The situation of the moisture field is slightly different. Even though the
model captures the evolution of relative humidity (RH) throughout 8 months, both mean and
median RH have ~10% bias in the model. In particular, the biases become more severe when RH
values fall into the high humidity regime. The RH frequency within the 90-100% range is about



two times higher in SCAM6 than Observation. A comparison of specific humidity (SH) shows that
SCAM6 overpredicts SH by 11.8%, indicating that the RH bias stems from the absolute moisture
bias, instead of temperature bias. It can be explained by the fact that temperature field is relaxed
to the input as an additional constraint, while SH is predicted as a fully prognostic variable in SCM.
We will examine the potential impact of moisture uncertainty in the large-scale forcing data on the
cloud property simulation through sensitivity, and the results will be discussed below.
**3.2 Cloud properties**
We first compare CLWC and RLWC over time and altitude dimensions between SCAM6
simulations and ARM radar-lidar-MWR retrievals (Figure 2a-d). The simulated CLWC values in
both time and altitude are generally consistent with the ARM retrievals. More specifically, SCAM6
can capture those thick clouds in early November and middle December due to the prevalent
frontal systems during that time of the year. However, some high CLWC values are not reproduced
in the model. Similarly, the temporal evolution of simulated RLWC agrees with the retrievals as
demonstrated in Figure 2c-d, however, their magnitudes are much lower than the retrievals. The
relatively coarse vertical resolution near the PBL is discernable from the discretized cloud vertical
distribution in the model simulations (Fig. 2a, c). However, the vertical development of different
cloud types (stratus, stratocumulus, and cumulus) and their transitions are generally reproduced by
SCAM6. When cumulus occurs with cloud top height greater that 2000 m, the model can always
capture them. Despite good agreement on clout top height, SCAM6 overpredicts CLWC and
RLWC frequency near the surface (< 200 m) compared to the observations. The statistics of cloud
macrophysics in Fig. 3 supports the analyses above. Cloud-top heights show good agreement
between SCAM6 simulation and Observation, with 8-month mean values of 1561 m and 1425 m,
respectively (Fig. 3f). It corroborates the notion that SCAM6 can capture the cloud type transition
relatively well. However, due to the lower cloud-base height in SCAM6, cloud physical thickness
is overestimated in the model. Even with the above biases in cloud macrophysics, the modeled
cloud mass center (CMC) height (mean cloud layer heights weighted by CLWC) is comparable to
the observed one (Fig. 3h).
A further comparison of 8-month surface precipitation rate in Fig. 2e and 2f shows that
SCAM6 can capture the heavy precipitation (>25 mm/day) under the large-scale forcing during
the winter season (Oct. to Jan.). However, the "too-frequent-drizzle" issue persists throughout the
8-month simulations. The frequency of light precipitation (< 2 mm/day) is more than 80% which



is rather unrealistic compared to the observations. The mean surface precipitation in SCAM6 is
overestimated by 30% compared to the rain gauge measurements during the whole 8-month period.

The statistical comparisons of cloud and drizzle microphysical properties in Fig. 3a-d

reveal that CLWC is overestimated by about 30%. Consequently, $r_c$ is slightly larger in the model,
and the bias becomes worse for those larger droplets ($r_c$ greater than 10 micron). Too large CLWC
fosters fast cloud to warm rain conversion, but the simulated RLWC values are smaller than the
retrievals, leading to too frequent surface precipitation mainly in the drizzle form.  Note that
retrieved RLWC from ground-based radar also bears with large uncertainty, as indicated by the
large error bar in Fig. 3c. Hence the real differences of RLWC between SCAM6 and Observation
remain hard to be quantified. Our analyses here include all 8-month simulation results and all types
of cloud during this time. In an additional analysis, we focus on the marine boundary layer (MBL)
stratiform cloud only but get quite similar cloud evaluation results. As shown in Fig. S1, then we
strengthen our selection criteria by only sampling consecutive cloud layers lasting more than 2
hours with the cloud top heights less than 3 km, the statistics of cloud micro- and macro- physical
properties do not differ significantly. It reflects the fact that over the ENA, MBL clouds are pre-
dominated during those seasons. In observation of the specific humidity bias against the
observations (Fig. 1), additional SCAM6 sensitivity test is conducted by perturbing moisture
content and the associated advection with a scaling factor of 0.85. Results show that the
distributions of simulated SH and RH only slightly shift towards the lower tail with smaller mean
values, which cannot correct their biases. Notably, despite the minor changes in the simulated
cloud and drizzle microphysics, the cloud-top height and thickness and the CMC simulations
perform noticeably better than the control simulation (Fig. S2). It suggests that the moisture fields
in the large-scale forcing exert larger impacts on simulated cloud structure and macrophysics than
the microphysics. In other words, cloud microphysical properties are strongly regulated by the
parameterizations, and less sensitive to the external forcing.

Driven by the same large-scale forcing, SCAM5 simulated cloud properties are quite

different from those by SCAM6. Instead of an overestimation in SCAM6, the SCAM5 simulated
CLWC exhibits an underestimation.  One possible reason is the change of formula for the
saturation vapor pressure in the MG2 cloud microphysics scheme (Gettelman and Morrison, 2015).
Previous single-column simulations for the MPACE case also show the larger LWC by MG2 than
MG1 (Gettelman et al., 2015). The good agreement of the mean $r_c$ in SCAM6 does not exist in the



SCAM5 simulations, and too many small cloud droplets (less than 6 micron) are present in
SCAM5, which are not found in either observation or SCAM6. RLWC in SCAM5 is still much
smaller than Observation, suffering the similar issue to SCAM6. Differing from SCAM6, SCAM5
overpredicts mean $r_{m,r}$ but underpredict mean $r_c$. The high bias in drizzle size but low bias in
drizzle amount in SCAM5 indicate that the sources of those biases can be different in the model.
The improvement of the cloud macrophysics from SCAM5 to SCAM6 is more evident than that
of microphysics. Too low cloud-base height and cloud-top height result in too thin cloud deck in
SCAM5. The cloud center mass is also systematically low in SCAM5. Overall speaking, the
updated cloud physics in CAM6 help improve many aspects of cloud simulations, but the drizzle
issues still linger on.
**3.3 Aerosols**

To probe the possible uncertainty sources for cloud droplet number concentration, vertical

profiles of aerosol and CCN number concentrations are compared between SCAM6 simulations
and aircraft in situ observations from 17 flights during the ACE-ENA field campaign (Fig. 5).
SCAM6 generally gets seasonality right, i.e., aerosol and CCN number concentrations are high in
summer and low in winter. The model also agrees with observations on the magnitude of
accumulation-mode aerosol concentration ($N_{ACC}$) and CCN concentration ($N_{CCN}$) during the
summer, which further leads to a reasonable comparison of $N_C$. The small bias of $N_C$ generally
follows the performance of $N_{CCN}$, i.e., high bias near the bottom while low bias near the top. One
intruiguing phenomenon during the summertime is that $N_{CCN}$ can be even higher than $N_{ACC}$, found
in both aircraft measurements and model simulations. The high $N_{CCN}$ occurs within the MBL
(<1000 m) in SCAM6. In contrast, measured $N_{CCN}$ in lower free troposphere (FT, 2000-2500 m)
is of the same magnitude with that within MBL, and FT $N_{CCN}$ is higher than $N_{ACC}$ in the
observations. A breakdown of aerosol number concentration budget in SCAM6 (Fig. S3) shows
that Aitken-model aerosols contribute to about 20% summertime and 45% wintertime total aerosol
numbers. In contrast, the coarse model aerosol number is only about 1% of the Aitken mode one.
Therefore, the large $N_{CCN}$ within the MBL in SCAM6 should be attributed to the efficient Aitken-
model aerosol activation near the cloud bottom in SCAM6. A further examination of aerosol
chemical composition in SCAM6 suggests that sulfate is the predominated aerosol species in the
Aitken model (Fig. S4). Understanding larger $N_{CCN}$ than $N_{ACC}$ in the lower FT in the observational
data is somewhat challenging, because coarse- and Aitken- mode aerosol number concentrations





was not measured during the IOP. However, previous study found that new particle formation
frequently occurs in the FT over the ENA, because of the sulfuric acids being elevated, especially
during summertime where the oceanic dimethyl sulfide (DMS) emissions are strong (Zawadowicz
et al., 2021). Previous back-trajectory analyses by Wang et al. (2020) suggest the long-range
transport of the fine-mode aerosols to the ENA site likely originates from the continental U.S.
Therefore, the oxidations of DMS, jointly with the long-range transported pollution, contribute to
the elevated Aitken-mode aerosol concentrations in the FT. Those Aitken-mode aerosols (e.g.,
DMS oxides and diluted continental pollutants) are found to be substantial contributors to the CCN
budget (Wang et al., 2021). The FT aerosols and CCN can be further entrained down to the MBL,
consistent with what is shown in Fig. 5. Note that SCAM6 predicts the "top-heavy" Aitken model
aerosol concentration profile, but it does not lead to the larger $N_{CCN}$ above the MBL. Hence, we
can only speculate that in the real atmosphere, there are significant Aitken mode aerosols that can
serve as CCN in the lower FT, but that is not the case in SCAM6. The above discussions reinforce
the notion that it is crucial to accurately simulate the long-range transport of aerosols over a remote
maritime region like ENA. And future investigation on how the aerosol activation processes are
being simulated in different model levels is warranted.

During the winter, $N_{ACC}$ is comparable between model and observation, while $N_{CCN}$ is

significantly overestimated from the surface to 2000 m altitude. Based on our analyses above for
the summer, we can infer that overestimated contribution from the Aitken mode to the CCN budget
also exists in winter. Moreover, it is non-negligible that the stronger convective activities due to
the frequent frontal passages during wintertime also likely result in the stronger activation of
Aitken mode aerosol. In contrast, the modeled $N_C$ shows surprisingly good agreement with
observations, despite the overestimated $N_{CCN}$. One plausible reason is the canceling effect from
the overestimated droplet size in the model (Fig. 3b). Larger cloud droplets facilitate the
autoconversion and accretion processes, and in turn, efficiently deplete cloud droplets (Zheng et
al., 2022b), keeping the observed $N_C$ at a comparable level with the model simulation.
**4. Impacts of new observation-constrained warm rain parameterizations**

To explore the possible sources of biases in simulated drizzle and LWC, we employ a

retuned KK scheme (Dong et al., 2021, thereafter as D21-KK) that explicitly links the
autoconversion and accretion rates with mass mean cloud droplet radius ($r_{m,c}$). The original
KK2000 scheme is expressed as below:




$$R_{auto}(Z) = \left(\frac{\partial q_r}{\partial t}\right)_{auto} = A\, q_c^{a1}(Z)N_c^{a2},$$ (1)

and,

$$R_{accr}(Z) = \left(\frac{\partial q_r}{\partial t}\right)_{accr} = B\left(q_c(Z)q_r(Z)\right)^b,$$ (2)

where A = 1350, $a1$ = 2.47, and $a2$ = -1.79 in CAM5. CAM6 aims to reduce the
autoconversion dependency on the $N_c$, so $a2$ and A are set as -1.1 and 13.5, respectively, with $a2$
unchanged. In D21-KK, both autoconversion and accretion rates are further aware of the vertical
variations of $r_c$, so the constant A and B are replaced as a function of $r_c$:

$$R'_{auto}(Z) = \frac{RLWC(Z)}{\int \rho_{air}P_r(Z)dt}R_{auto}(Z) = A'(Z)q_c^{2.47}(Z)N_c^{-1.79},$$ (3)

and,

$$R'_{accr}(Z) = \frac{RLWC(Z)}{\int \rho_{air}P_r(Z)dt}R_{accr}(Z) = B'(Z)\left(q_c(Z)q_r(Z)\right)^{1.15},$$ (4)

where A' and B' are further parameterized in CAM5 as:

$$A'(Z) = 121683exp\left(-0.528\, r_{m,c}(Z)\right) + 364,$$ (5)

and,

$$B'(Z) = 632exp\left(-24.5\frac{r_{m,c}(Z)}{r_{m,r}(Z)}\right) + 51.$$ (6)

Dong et al. (2021) showed that this set of new parameterizations in CAM5 help alleviate
the long-lasting issue in the climate models, e.g., "too frequent and too light precipitation", on the
global scale. When we apply the same set of parameterizations in SCAM5 over the ENA (referred
to as SCAM5$_{D21}$), we find similar improvements on cloud and precipitation properties. As shown
in Fig. 6, CLWC in SCAM5$_{D21}$ is elevated due to the less efficient autoconversion scheme, and the
simulated CLWC values agree better with the ARM retrievals compared with original SCAM5. $r_c$
is also enlarged in SCAM5$_{D21}$, becoming more consistent with retrievals. The mass median radius
of raindrops $r_{m,r}$ are reduced slightly, while there is no significant change in RLWC in SCAM5$_{D21}$.
Because of the improved cloud microphysical properties, cloud macrophysics also match up better
with observations. Cloud base height, cloud top height, and cloud mass center height (Fig. 6e-h)
are all improved to some extent in SCAM5$_{D21}$ simulations. These comparisons are encouraging,
indicating that the D21-KK new warm parameterizations in SCAM5 make significant
improvements on the simulated MBL cloud and drizzle properties.



341 Different from CAM5 microphysics, CAM6 starts to introduce sub-grid cloud variations

342 (Zhang et al., 2020) and re-tuned the parameters in the KK2000 scheme. One direct consequence

343 is that cloud LWC has been changed from underestimation to overestimation (Fig. 7a). Therefore,

344 an even slower autoconversion process with the new D21-KK scheme cannot further benefit the

345 warm rain processes in CAM6. As expected, SCAM6$_{D21}$ does not exhibit improvement in

346 simulating both cloud microphysics and macrophysics (Fig. 7). Distinctive sensitivities to the same

347 microphysical parameter modification under different physics packages poses a challenge on

348 model improvement through only updating a certain set of parameterizations.

**5. Assessing aerosol indirect effects under the single-column frameworks**

350 Aerosol indirect effects, especially the second indirect effect concerning the liquid water

351 path change, was reported to be over-predicted in CAM5 when simulating the aerosol

352 perturbations, such as volcano eruptions, on the low clouds (Malavelle et al., 2017). Here we assess

353 the aerosol first and second indirect effects of CAM6 over the ENA under the single-column

354 framework. To perturb the CCN budget, we choose to modify the accumulation-mode aerosols in

355 their initial conditions.  As the aerosol relaxation is on, such a perturbation is expected to constantly

356 impact the aerosol field during the integrations. Considering the relatively low background aerosol

357 concentration, the change in aerosol direct effect on the clear-sky radiation fluxes can be ignored

358 in this setup. Both aerosol number and mass concentrations in the accumulation mode are enlarged

359 by a factor of 2, the results are labeled as S6$_{pAero}$ and are compared with the original SCAM6

360 simulations (Fig. 8). With such an aerosol perturbation, $N_{CCN}$ within MBL (< 1km) is increased

361 from 112.5 to 175.8 cm$^{-3}$, corresponding to a 56% enhancement. Similarly, CCN in the lower FT

362 and upper MBL (1-3 km) increased by 61%. Aerosol first and second indirect effects are evident

363 in SCAM6, as reduced $r_c$ and increased LWC are both found in the perturbed experiment. We

364 further quantify the droplet size susceptibility and cloud water susceptibility with respect to MBL

365 CCN changes by $\frac{\partial ln(r_c)}{\partial ln(N_{CCN})}$ and $\frac{\partial ln(CLWC)}{\partial ln(N_{CCN})}$, respectively. The SCAM6 simulated droplet size

366 susceptibility is –0.2, close to the LES simulated range from –0.22 to –0.25 and the upper bound

367 of the observed range over ENA (Wang et al., 2020; Zheng et al., 2022a). The SCAM6 simulated

368 cloud water susceptibility is +0.19 which also falls into the LES prediction (+0.18 to +0.30). Those

369 results suggest that the newly introduced sub-grid cloud variabilities in SCAM6 can account for

370 the aerosol indirect effects at a reasonable level. Mean surface precipitation amount shows very

371 small responses to CCN perturbation (less than 2%), because convective precipitation in early



winter dominates the study period while deep convective parameterization in SCAM6 is still
unlinked with cloud microphysics and unaware of CCN effects so far. Cloud top height ($Z_T$) shows
an increase with higher CCN concentration (Fig. 8f), likely due to the enhanced latent heat release
following the elevated condensational rate.

**6. Conclusion and Discussion**

The single-column versions of NCAR CAM5 and CAM6 are employed to simulate marine

boundary-layer cloud and aerosol properties over the eastern North Atlantic during the ACE-ENA
field campaign and to assess the uncertainty in cloud microphysical parameterizations. 3-hourly
large-scale forcing data are derived from the systematic measurements of atmospheric states
during the 8-month IOP. SCAM6 well reproduces the temperature field but overestimates specific
and relative humidity by about 10%, especially for those near-cloud grid points.  Our moisture
adjustment simulation suggests that moisture variables in the large-scale forcing exert larger
impacts on simulated cloud structures than cloud microphysics. It further implies cloud
microphysical properties are strongly regulated by the parameterizations, and less sensitive to the
external forcing. Cloud frequency and transition between different types show good agreement
between SCM and observation. Cloud property simulations are generally improved from SCAM5
to SCAM6, in terms of droplet effective radius, cloud top height, and cloud thickness. However,
there are some common issues with warm precipitation in those two models, including too small
rainwater content and too frequent surface light precipitation. To probe the possible contributions
from the warm cloud parameterization to those drizzle biases, we implement the recalibrated
autoconversion and accretion processes in the KK scheme of SCAM5 and SCAM6 that explicitly
consider vertical variations of droplet size. This updated scheme tends to improve CLWC and $r_c$
in SCAM5 as well as $r_{m,r}$ , but does not significantly alleviate the drizzle problem. The
improvement is absent in SCAM6, likely because sub-grid variations of cloud properties have been
introduced in CAM6 cloud microphysics (especially for the autoconversion parameterization),
suppressing the KK scheme sensitivity to other factors. Further study is warranted to test whether
the same warm rain precipitation sensitivity holds for different cases using SCM5/6.

Aerosol simulations in SCAM6 are evaluated against the aircraft measurements during the

ACE-ENA. SCAM6 agrees with observations on the magnitude of concentration of accumulation-
mode aerosol, CCN, and cloud droplets during the summer, while $N_{CCN}$ is significantly biased high
from the surface to 2000 m in altitude during the winter. Aerosol budget analyses show that in



SCAM6, long-range transport provides too many Aitken-mode sulfates that entrain into the MBL
and can grow to CCN-size particles consequently. We further quantify aerosol indirect effects by
perturbing accumulation-mode aerosol concentrations in the model. SCAM6 predicted cloud water
and droplet size susceptibilities line up with the classic CCN effects, i.e., reduced droplet size but
enhanced liquid water content under the high CCN scenario. The magnitudes of the cloud water
and droplet size susceptibilities are also close to the LES simulations conducted for the selected
cases during the ACE-ENA.

The present study provides new insight of model biases in aerosol and warm cloud

simulations in the NCAR CAM models. Different from the previous evaluations of a full model
run with potential large biases propagated from modeled large-scale conditions, the model biases
discussed here, especially the drizzle property issue, should be adequately addressed in the future
development of CAM. The existing progress of predicted cloud properties and aerosol effects is
clearly demonstrated under the single-column framework in this study.

**Code availability**
The    code    of    CESM    model    used    in    this    study    is    available    at
https://www.cesm.ucar.edu/models/cesm2/release_download.html.

**Data availability**
All the CESM model simulation input and output used for this research can be downloaded from
the website at http://web.gps.caltech.edu/~yzw/share/Wang-2023-SCM. The aircraft and ground-
based measurements used in this study were obtained from the Atmospheric Radiation
Measurement (ARM) Program sponsored by the U.S. Department of Energy (DOE) Office of
Energy Research, Office of Health and Environmental Research, and Environmental Sciences
Division. The data can be downloaded from http://www.archive.arm.gov/.

**Competing interests**
Yuan Wang is a member of the editorial board of Atmospheric Chemistry and Physics.

**Acknowledgement**
This study was primarily supported by the collaborative NSF grant (Award No. AGS-2031751,



2031750). We thank the instrument mentors of the instruments and the individuals collecting
measurements during the ACE-ENA field campaign. We also acknowledge high-performance
computing support from NCAR Cheyenne.  All requests for materials in this paper should be
addressed to Yuan Wang (yuanwang@purdue.edu).



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



**Table 1**. Single-column numerical experiment design.

| Model Physics | Experiment Name | Experiment Description |
|---|---|---|
| CAM6 | Ctrl | Default model setup and forcing data |
| | D21 | Using recalibrated warm rain parameterizations based on Dong et al. (2021) |
| | pAero | Scale up aerosol number and mass concentrations in the accumulation mode by a factor of 2 in the initial condition |
| | ForcingQ_Adj | Adjust specific humidity state variable and related tendency terms by a factor of 0.85 |
| CAM5 | Ctrl | Default model setup and forcing data |
| | D21 | Using recalibrated warm rain parameterizations based on Dong et al. (2021) |





**Figures**

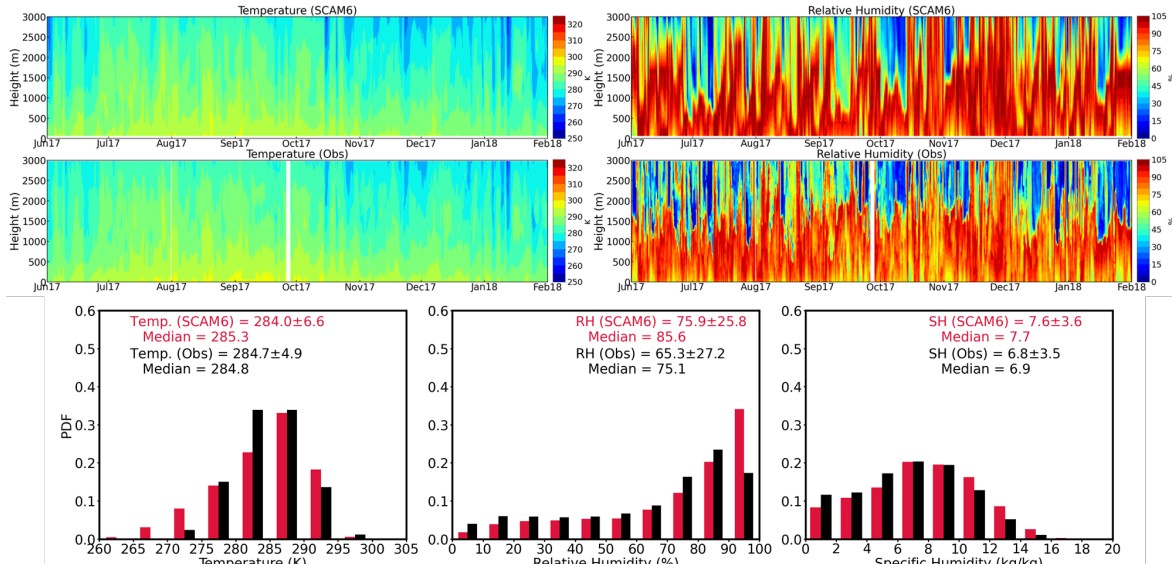

**Figure 1**. Comparisons of meteorological conditions between SCAM6 simulations and ARM Interpolated Sonde (INTERPSONDE) soundings. Upper panels: Time series of air temperature (left) and relative humidity (right) from SCAM6 (top) and ARM-ENA observations (bottom). Lower panels: SCAM6 (red) simulated air temperature, relative humidity (RH), specific humidity (SH) within 3 km, in comparison with the ARM-ENA observation (black).



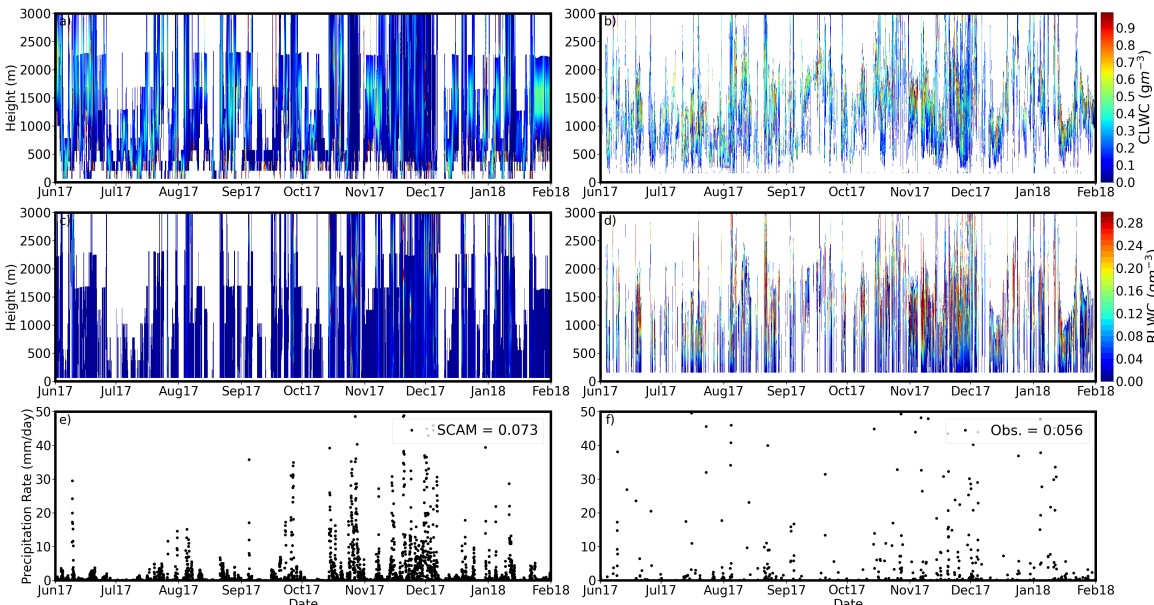

**Figure 2.** Time series of the cloud liquid water contents (CLWC, top panels), rain liquid water contents (RLWC, middle panels) and surface precipitation (bottom panels) from the SCAM6 simulations (left column) and the ARM-ENA retrievals and observations (right column).

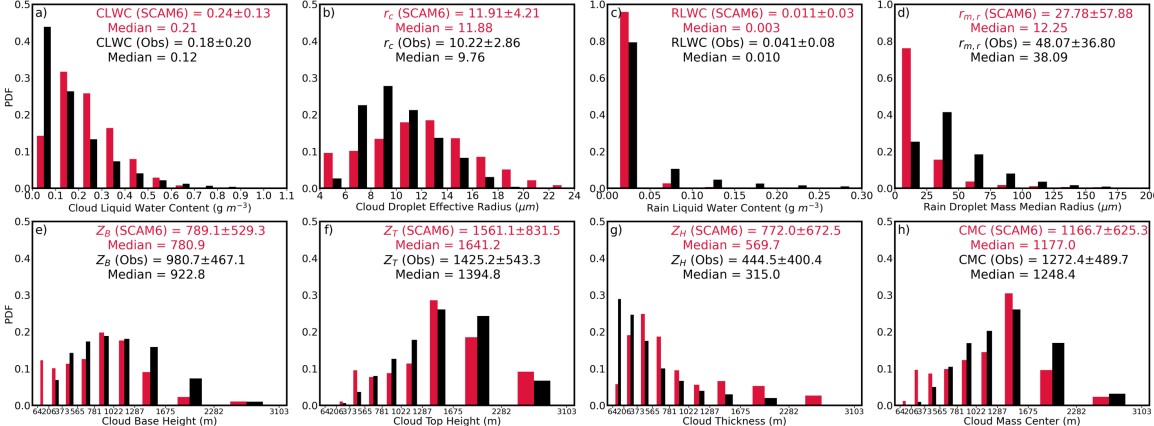

**Figure 3**. Probability distribution functions (PDFs), mean, standard deviation, and median values of cloud and rain microphysics, and cloud macrophysics simulated from SCAM6 (red) and observed/retrieved from ground-based remote sensors (black). (a) Cloud liquid water content, CLWC; (b) Cloud droplet effective radius, $r_c$; (c) Rain liquid water content, RLWC; (d) Rain droplet mass median radius, $r_{m,r}$; (e) Cloud base height, $Z_B$; (f) Cloud top height, $Z_T$; (g) Cloud thickness, $Z_H$ and (h) Cloud mass center.





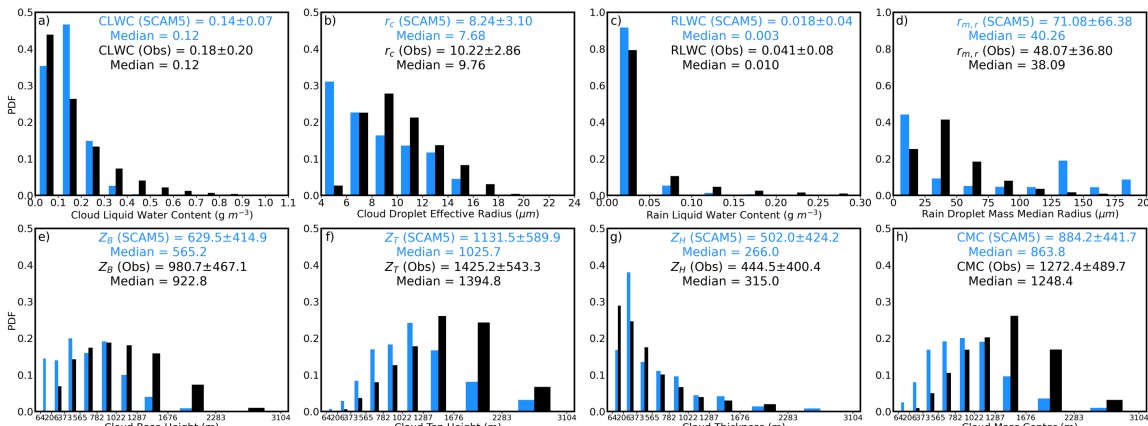

**Figure 4**. Same as Fig 3, except for SCAM5 (blue).

597

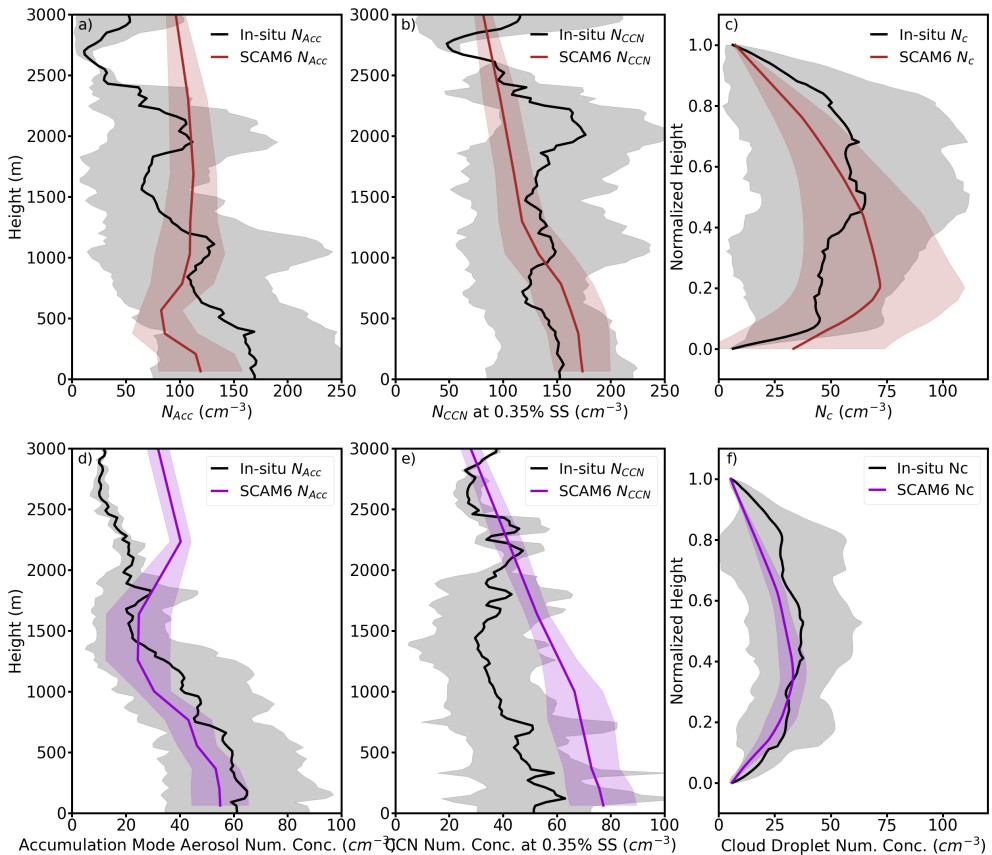

**Figure 5**. Vertical profiles of accumulation mode aerosol ($N_{ACC}$) (a, d); CCN concentration ($N_{CCN}$) at 0.35% supersaturation (b, e) during interstitial conditions, and Cloud droplet number concentration ($N_C$) at normalized height (c, f, 0 is cloud base, 1 is cloud top) for cloudy samples. For SCAM6 simulations (brown and purple) and aircraft in-situ measurement (black), during the Summer (top panels) and Winter (bottom panels) ACE-ENA IOPs. The shaded areas denote the standard deviation at each level. The SCAM6 simulations are selected within each time duration of the aircraft cases.

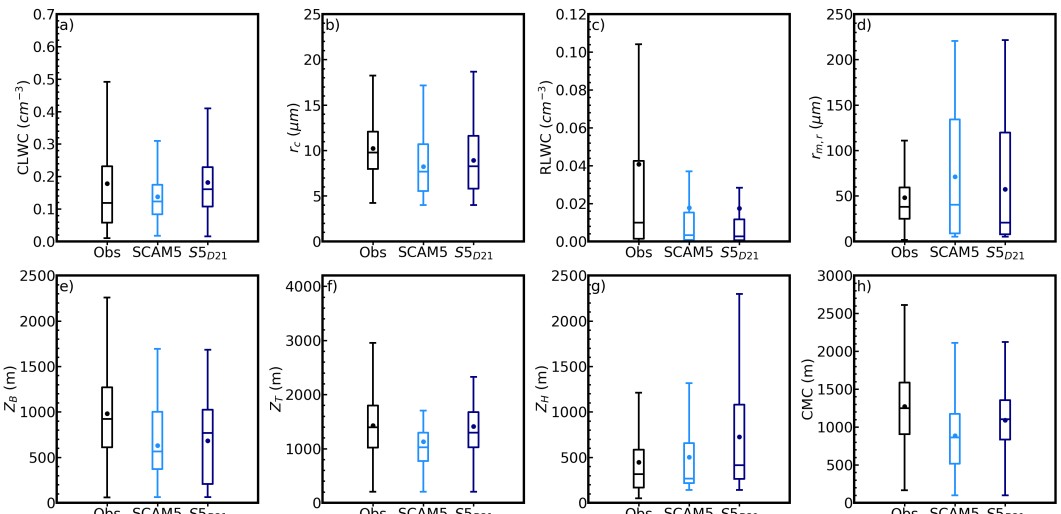

607

**Figure 6**. Comparisons of cloud and rain microphysics, and cloud macrophysics between observations (black), SCAM5 (blue) and SCAM5 with Dong2021 parameterization ($SCAM5_{D21}$, dark blue). (a) $CLWC$, (b) $r_c$, (c)$RLWC$, (d) $r_{m,d}$, (e) $Z_B$, (f) $Z_T$, (g) $Z_{H,}$ and (h) Cloud mass center. Dots represent the mean values, and the bars from bottom to top represent 10%, 25%, 50%, 75%, and 90% values, respectively.




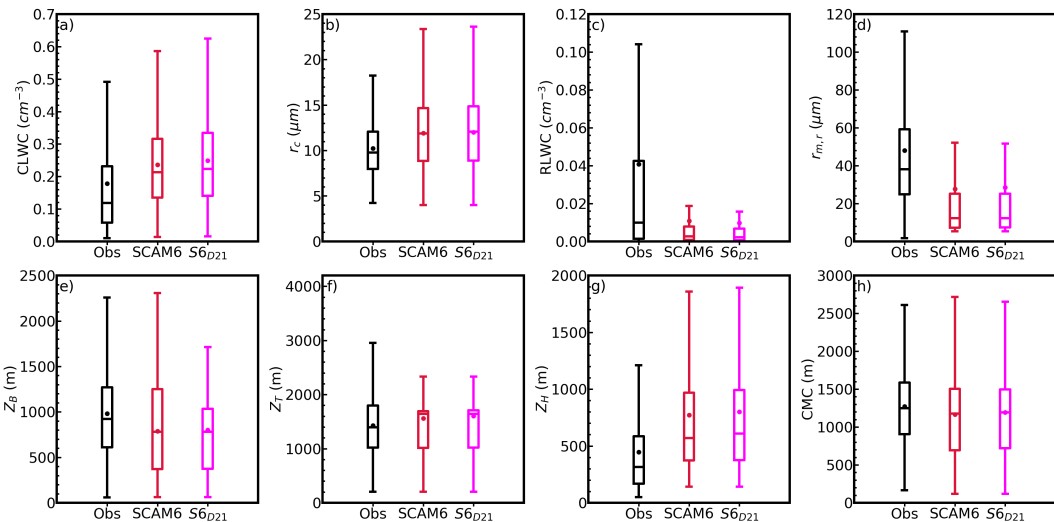


**Figure 7**. Same as Fig 6, except for SCAM6 (red), and SCAM6 with Dong2021 parameterization

($SCAM6_{D21}$, pink).





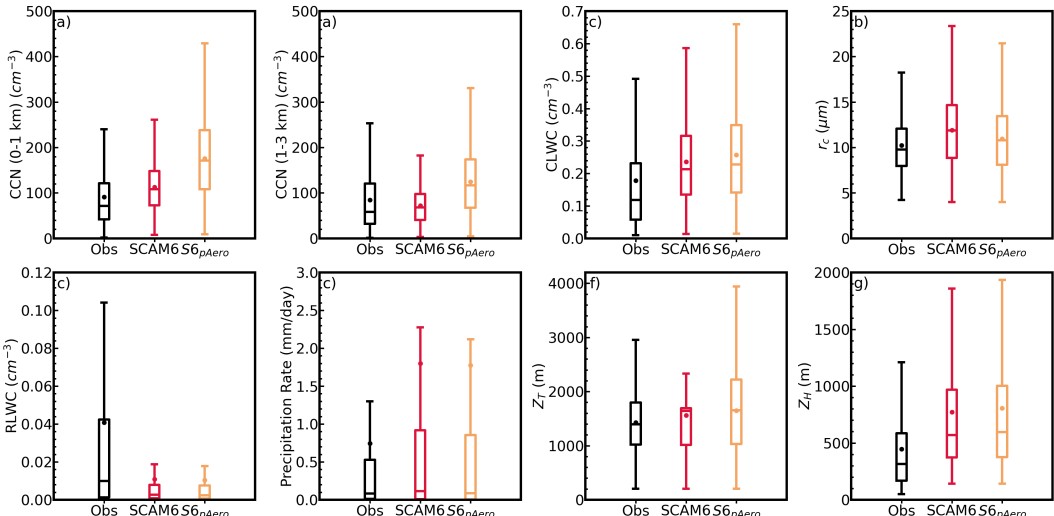

**Figure 8**. Aerosol and cloud properties simulated from control (red) and aerosol-perturbing
experiments (pAero, orange) by SCAM6 and comparison to observations. The observed CCN at
0.35% SS are averaged from the selected aircraft measurements during the ACE-ENA.