# Peer review of "Insights of warm cloud biases in CAM5 and CAM6 from the single-column modeling"

_EGUsphere, 2023_

## Referee Comment (RC1)

Review of "Insights of warm cloud biases in CAM5 and CAM6 from the single-column modeling framework and ACE-ENA observations" by Wang et al.

In this study, model biases in aerosol and warm cloud simulations are examined in two versions of the NCAR CAM model using a single-column model framework (SCAM5 and SCAM6) for the ACE-ENA field campaign. The authors analyze differences between simulated cloud and aerosol properties and ACE-ENA observational data.

The paper is well organized and written, but lacks clarity and important information. My general comments reflect this issue.

Major comments:
1. SCAM5/6 configuration and ACE-ENA case setup:
   a. To enhance the comprehension and reproducibility of the study, it would be beneficial to include more comprehensive details on the configuration and setup of SCAM5/6 simulations. Specifically, but not exclusively, the manuscript could provide information on which parameterizations were employed, which large-scale forcings were included, the evolution of input thermodynamic profiles over time (stationary or not?), the number of vertical levels used, and the model time step. Additionally, it may be useful to explicitly mention that the moisture field evolves freely, assuming it does so based on L192.
   b. The retuned KK scheme is mentioned for the first time in section 4. To enhance the paper's clarity for all readers, including those who are not experts on cloud microphysics parameterizations, I suggest introducing the retuned KK scheme in the Methodology section (the mathematical description can stay in section 4) along with a brief description of the default cloud microphysics parameterizations of SCAM5 and 6.

2. I have a few related comments in section 3.1:
   a. Although the simulated median and mean temperature values agree well with the observations, the temperature PDF suggests that this is partially due to a "canceling effect" from the lower/higher bins relative to the middle ones., i.e., the simulated values over the temperature "extremes" (lower and higher bins) are larger than the observed ones, but in the middle bins, the observed values surpass the simulated ones (i.e., bins between 280 and 290 K). All to say that the temperature PDF doesn't fully support the sentence in L184–185.
   b. The authors conclude that the specific humidity bias of SCAM6 arises from the model moisture bias rather than the temperature bias (L190–191), partially because the temperature field is nudged towards the initial conditions (L133–134). While this is true, the nudging time-scale for the PBL is rather long (close to 10 days) which reduces the nudged impact on the PBL's evolution. In addition, the temperature PDF indicates a moderate bias in simulated temperature in the lowest bins, i.e., between 265 to 275 K. Hence, while not entirely disagreeing with

L190–191, the results in Figure 1 do not rule out the possibility that the temperature field also contributes to the RH bias.

c. The limits of the temperature plots are unnecessarily large. To improve the clarity of the temperature field, I suggest reducing the upper and lower limits to 260–300 K; this range is also consistent with the temperature PDF below.

Please comment on these and adjust section 3.1 and figure 1 accordingly.

Minor comments:

P4 L93–94: I can't find the reference to Wang et al. (2016) and Zhang et al. (2020) in the References list.

P4 L93–94: Please consider adding at least one more reference per reference set, and add "e.g.," before each reference set since there are too many available references to include all.

P4 L111–112: Please define explicitly the acronym MAM.

P6 L150: Please include the estimated median uncertainties also for $N_c$ and $CLWC$.

P7 L167: The sentence "To make better… only selects the research flights with an "L" shape pattern center at the ARM-ENA site" may require additional context for readers who are not familiar with the flight sampling configuration used during ACE-ENA and its rationale. How does this pattern help improve comparisons between observations and simulations?

P10 L256: Could the authors provide more information on the physics used in the SCAM5 version in this study? In CAM6, CLUBB is responsible to diagnose the cloud macrophysical properties. To improve clarity, it would be helpful to include further information about the differences in the physics between the SCAM5 and SCAM6 versions used here; this ties in with my "Major comment 1". This should probably go in section 2.1.

P10 L264: Could you please confirm whether these in-situ profiles represent an average of data from the 17 flights? Also, could you clarify what the SCAM6 profiles correspond to? Are they averages of the 17-flight time-stamps, or do they represent something else?

P11 L308: In section 4, the results show that the retuned KK scheme improves cloud micro- and macrophysics in SCAM5, but "as expected" it doesn't lead to improvements in SCAM6 relative to the default MG2 (if I understood correctly). Thus, I was left at the end of section 4, questioning its purpose. I'm not suggesting removing it, but consider clarifying what this section adds to the paper.

P16 L423: The website link to where the data is stored is currently not working.

Figure 5: This is just a suggestion: Use the x-axis labels on only the bottom row or use the same labels on both rows. Currently, the bottom and upper rows have different x-axis labels even though they represent the same variable, which is a bit inconsistent.

---

## Author Response (AR1)

Review of "Insights of warm cloud biases in CAM5 and CAM6 from the single-column modeling framework and ACE-ENA observations" by Wang et al.

In this study, model biases in aerosol and warm cloud simulations are examined in two versions of the NCAR CAM model using a single-column model framework (SCAM5 and SCAM6) for the ACE-ENA field campaign. The authors analyze differences between simulated cloud and aerosol properties and ACE-ENA observational data. The paper is well organized and written, but lacks clarity and important information. My general comments reflect this issue.

We appreciate the reviewer's detailed comments and constructive suggestions. We have carefully revised the manuscript according to these valuable comments. Point-to-point responses are provided below. The reviewers' comments are in black, our responses are in blue, and the quotes from our manuscript are in italics.

Major comments:

1. SCAM5/6 configuration and ACE-ENA case setup:
    To enhance the comprehension and reproducibility of the study, it would be beneficial to include more comprehensive details on the configuration and setup of SCAM5/6 simulations. Specifically, but not exclusively, the manuscript could provide information on which parameterizations were employed, which large-scale forcings were included, the evolution of input thermodynamic profiles over time (stationary or not?), the number of vertical levels used, and the model time step. Additionally, it may be useful to explicitly mention that the moisture field evolves freely, assuming it does so based on L192.

As suggested, we have now provided more descriptions of model physics and configuration. A new Table 1 has been added to summarize the key physical parameterizations in CAM5 and CAM6 related with warm clouds:

| Model Physics | CAM5 | CAM6 |
|---|---|---|
| Cloud Microphysics | MG1 (*Morrison and Gettelman*, 2008) with KK scheme for warm rain processes. | MG2 with retuned autoconversion, explicit sub-grid variance of cloud, and prognostic rain and snow (*Morrison and Gettelman*, 2015) |
| Stratiform Macrophysics | The Park scheme (*Park et al.*, 2014) | The Cloud Layers Unified By Binormals (CLUBB), a prognostic moist turbulence scheme that unifies the representation of boundary layer, shallow convection, and stratiform macrophysics (*Golaz and Larson*, 2002) |
| PBL and shallow convection scheme | The University of Washington scheme (*Park and Bretherton*, 2009) | |
| Aerosol | 3-mode Modal Aerosol Module (MAM3, *Ghan et al.*, 2011) | 4-mode Modal Aerosol Module (MAM4) with a new "fresh-BC" mode (*Liu et al.*, 2016) |

We have also clarified in the revised manuscript that the large-scale forcing data include temperature (T) and moisture (q), their horizontal and vertical advection, surface sensible and latent heat fluxes, U and V winds, large-scale vertical motion/velocity, TOA/surface radiation fluxes, etc. The data have evolution of thermodynamic profiles over time with 3-hr intervals. The moisture field are subject to both large-scale forcing data and model physics. CAM6 model has 32 vertical levels from the surface to 2 hPa (about 45 km), while CAM5 have 30 levels. Both two models has a time step of 30 minutes, while CAM6 uses sub-stepping for microphysical processes.

The retuned KK scheme is mentioned for the first time in section 4. To enhance the paper's clarity for all readers, including those who are not experts on cloud microphysics parameterizations, I suggest introducing the retuned KK scheme in the Methodology section (the mathematical description can stay in section 4) along with a brief description of the default cloud microphysics parameterizations of SCAM5 and 6.

To describe the KK scheme in CAM5/6 and our modification on it, we have now added a new section 2.2 with all the relevant equations and parameters. The discussions on MG1/2 cloud microphysics have now been expanded in the section 2.1 and new Table 1.

I have a few related comments in section 3.1:
Although the simulated median and mean temperature values agree well with the observations, the temperature PDF suggests that this is partially due to a "canceling effect" from the lower/higher bins relative to the middle ones., i.e., the simulated values over the temperature "extremes" (lower and higher bins) are larger than the observed ones, but in the middle bins, the observed values surpass the simulated ones (i.e., bins between 280 and 290 K). All to say that the temperature PDF doesn't fully support the sentence in L184–185.
The authors conclude that the specific humidity bias of SCAM6 arises from the model moisture bias rather than the temperature bias (L190–191), partially because the temperature field is nudged towards the initial conditions (L133–134). While this is true, the nudging time-scale for the PBL is rather long (close to 10 days) which reduces the nudged impact on the PBL's evolution. In addition, the temperature PDF indicates a moderate bias in simulated temperature in the lowest bins, i.e., between 265 to 275 K. Hence, while not entirely disagreeing with L190–191, the results in Figure 1 do not rule out the possibility that the temperature field also contributes to the RH bias.
The limits of the temperature plots are unnecessarily large. To improve the clarity of the temperature field, I suggest reducing the upper and lower limits to 260–300 K; this range is also consistent with the temperature PDF below. Please comment on these and adjust section 3.1 and Figure 1 accordingly.

We agree with the reviewer's comment that the temperature bias contribution cannot be ruled out in the previous analyses. We revised our discussions and Fig. 1 as suggested. To better attribute the RH high bias in the model, we have now analyzed the T and SH for the points with RH larger than 90% where the largest RH biases are found. The statistics below shows the discrepancy of SH is still larger than that of T, implying SH may contributing more to the RH biases in this regime by comparing the means, even though the uneven T biases remain. The figure below is added a new SI Figure.

[Figure]

Below is the new Fig. 1 modified as suggested:

[Figure]

Minor comments:

P4 L93–94: I can't find the reference to Wang et al. (2016) and Zhang et al. (2020) in the References list.

We have now updated the two references' information as below.

*Wang, Y., P.-L. Ma, J. Jiang, H. Su and P. Rasch, Towards Reconciling the Influence of Atmospheric Aerosols and Greenhouse Gases on Light Precipitation Changes in Eastern China, J. Geophys. Res. Atmos. 121(10), 5878–5887, 2016.*

*Zhang, Z., Song, H., Ma, P.-L., Larson, V. E., Wang, M., Dong, X., and Wang, J.: Subgrid variations of the cloud water and droplet number concentration over the tropical ocean: satellite observations and implications for warm rain simulations in climate models, Atmos. Chem. Phys., 19, 1077–1096, https://doi.org/10.5194/acp-19-1077-2019, 2019.*

P4 L93–94: Please consider adding at least one more reference per reference set, and add "e.g.," before each reference set since there are too many available references to include all.

Good suggestion. We have revised as suggested.

P4 L111–112: Please define explicitly the acronym MAM.

Defined as suggested.

P6 L150: Please include the estimated median uncertainties also for Nc and CLWC.

We have now clarified that the cloud and drizzle microphysical retrievals were validated by the aircraft in-situ measurements from ACE-ENA field campaign, with the estimated median uncertainties of ~15% for $r_c$; ~30% for $r_{m,r}$; ~30% for $N_c$ and CLWC, and ~50% for $N_r$ and $RLWC$ (Wu et al., 2020).

P7 L167: The sentence "To make better… only selects the research flights with an "L" shape pattern center at the ARM-ENA site" may require additional context for readers who are not familiar with the flight sampling configuration used during ACE-ENA and its rationale. How does this pattern help improve comparisons between observations and simulations?

We have now clarified that to facilitate the model-observation comparisons, we selected only those research flights that followed a horizontal track within one grid size of the CAM models (1.25° longitude and 0.9° latitude), centered on the ARM-ENA site. Also to meet the criteria for comparison with SCAM6, each aircraft case must include comprehensive vertical sampling of cloud and aerosol within the specified time period. Table S1 lists the dates and time periods of the selected flights.

P10 L256: Could the authors provide more information on the physics used in the SCAM5 version in this study? In CAM6, CLUBB is responsible to diagnose the cloud macrophysical properties. To improve clarity, it would be helpful to include further information about the differences in the physics between the SCAM5 and SCAM6 versions used here; this ties in with my "Major comment 1". This should probably go in section 2.1.

We have now added a new Table 1 to summarize the relevant physical parameterizations used by CAM5 and CAM6.

P10 L264: Could you please confirm whether these in-situ profiles represent an average of data from the 17 flights? Also, could you clarify what the SCAM6 profiles correspond to? Are they averages of the 17-flight time-stamps, or do they represent something else?

We have now clarified that the in-situ profiles represent the average of data collected during 12 flights and 5 flights selected during the summer and winter IOPs, respectively. Those flights are chosen because they overlap with our model simulation period and their track near the Azores

islands where are SCM is set up. The SCAM6 profiles correspond to the averages within the 17-flight time-stamps. The information of 17 flights is now provided in the new Table S1.

P11 L308: In section 4, the results show that the retuned KK scheme improves cloud micro- and macrophysics in SCAM5, but "as expected" it doesn't lead to improvements in SCAM6 relative to the default MG2 (if I understood correctly). Thus, I was left at the end of section 4, questioning its purpose. I'm not suggesting removing it, but consider clarifying what this section adds to the paper.

We are sorry for the confusion arising from our unclear statements and insufficient description. In fact, our SCAM6 D21 experiment also recalibrated autoconversion parameterization using the method to similar Dong et al. (2021), not the identical parameters. The accretion parameterization remains the same, as CAM6 did not retune its parameters. We have not added in Section 4 as below: *"We did the similar recalibration for CAM6 autoconversion processes, and the corresponding A' is parameterized as:*

$$A'(Z) = 3359 \times exp\left(-0.721\, r_{m,c}(Z)\right) + 8, \hspace{3cm} (7)"$$

Therefore, it was expected that this recalibration on CAM6 would do the same improvement on the warm rain rates. However, the results deny this hypothesis and the likely cause is the newly introduced sub-grid cloud variations in CAM6.

P16 L423: The website link to where the data is stored is currently not working.

The problem has been fixed now.

Figure 5: This is just a suggestion: Use the x-axis labels on only the bottom row or use the same labels on both rows. Currently, the bottom and upper rows have different x-axis labels even though they represent the same variable, which is a bit inconsistent.

Thanks for pointing this out. Now the x-axis labels have been fixed and consistent.

Review of "Insights of warm cloud biases in CAM5 and CAM6 from the single-column modeling framework and ACE-ENA observations" by Wang et al.

This manuscript presents a study using the single column configurations of NCAR CAM5 and CAM6 to simulate marine boundary-layer cloud and aerosol properties over the eastern North Atlantic during the ACE-ENA field campaign. The authors further accessed the uncertainty in cloud microphysics parameterization.

The manuscript is clear in addressing scientific questions and well analyzes the results. The figures also strongly support the analysis from model results and observations. However, readers are hard to follow the main points in the current structure of the manuscript, especially for those unfamiliar with CAM5 and CAM6. Some general comments reflect my concern.

We appreciate the reviewer's detailed comments and constructive suggestions. We have carefully revised the manuscript according to these valuable comments. Point-to-point responses are provided below. The reviewers' comments are in black, our responses are in blue, and the quotes from our manuscript are in italics.

**General comments:**

1. The title of the manuscript highlights the main discussion focusing on the warm cloud biases in CAM5 and CAM6, but the authors did not clearly point out the differences between CAM5 and CAM6. A table could help readers compare the main differences between the two models.

As suggested, we have now created a new Table 1 to summarize the physical parameterizations relevant with warm rain processes.

2. Section 3 is to evaluate SCAM6 using ACE-ENA observations. Again, I do not know whether I should expect that SCAM5 and SCAM6 have similar results. The section title should be changed since the authors added many contents of SCAM5. The same figures as Fig. 1 and Fig. 2 should be present in the manuscript or supplementary for SCAM5.

We have changed the subtitle for Section 3 as "Evaluation of SCAM using ACE-ENA observations". As suggested, we have also provided the SCAM5 evaluation figures similar to Figs. 1 and 2. Since the simulations of the meteorological fields by SCAM5 largely resemble those by SCAM6, we put those two figures into our supplementary materials and discuss them in the main text. The similarity is expected as they are driven by the same large-scale forcing data.

[Figure]

**Figure S4**. The same with Fig. 1 but for SCAM5.

[Figure]

**Figure S5**. The same with Fig. 2 but for SCAM5.

3. Returned KK scheme (D21-KK) has improved autoconversion and accretion rates with mean cloud droplet radius. However, the turned coefficients are tested in CAM5 by Dong et al., 2021. It seems reasonable if D21-KK did not offer a better result in SCAM6 because those coefficients need to be returned for CAM6. Why did the authors think the results came from introducing sub-grid cloud variations in CAM6?

We are sorry for this confusion stemming from our unclear statements and insufficient description. In fact, our SCAM6 D21 experiment recalibrated autoconversion parameterization using the method to similar Dong et al. (2021), not the identical parameters. The accretion parameterization remains the same, as CAM6 did not retune its parameters. We have now added in the Section 2.2 as: *"CAM6 microphysics aims to reduce the autoconversion dependency on the $N_c$, so a2 and A are set as -1.1 and 13.5, respectively, with a2 unchanged. We did the same recalibration for CAM6 autoconversion processes, and the corresponding A' is parameterized as:*

$$A'(Z) = 3359 \times exp\left(-0.721\, r_{m,c}(Z)\right) + 8, \tag{7}$$

*Hence the updated autoconversion for CAM6 microphysics has the form as below:*

$$R'_{auto}(Z) = \frac{RLWC(Z)}{\int \rho_{air} P_r(Z) dt} R_{auto}(Z) = f_e\, A'(Z) q_c^{2.47}(Z) N_c^{-1.1}, \tag{8}$$

*Where $f_e$ represents an enhancement factor which is diagnosed from the CLUBB to account for sub-grid variabilities of cloud and rain."*

Therefore, it was expected that this recalibration on CAM6 would do the same improvement on the warm rain rates. However, the results deny this hypothesis and the likely cause is the newly introduced sub-grid cloud variations in CAM6.

4. The authors provided Table 1 in the manuscript but did not mention it in the text. All experiments, including those experiment names, are hard to follow in the manuscript.

To clarify our experiments, we have now added a new section 2.2 entitled "Numerical experiment design" on page 6 to detail the experiments and echo the Table 1.

**Specific comments:**

Lines 101-102: The two-moment cloud microphysical scheme is updated to version 2 (MG2; Gettelman and Morrisons, 2015) …

Revised as suggested.

Lines 156-157: Furthermore, the CLWC (RLWC) is scaled by the cloud (rain) fraction within …

Corrected as suggested.

Line 158: in-situ or in situ should be consistent throughout the paper.

We now use "in situ" throughout the paper.

Lines 167-168: Why choose the research flights with a "L" shape?

We have now clarified that to facilitate the model-observation comparisons, we selected only those research flights that followed a horizontal track within one grid size of the CAM models (1.25° longitude and 0.9° latitude), centered on the ARM-ENA site. Also, to meet the criteria for comparison with SCAM6, each aircraft case must include comprehensive vertical sampling of cloud and aerosol within the specified time period. Table S1 lists the dates and time periods of the selected flights.

Lines 251-260: The authors did not mention fig. 4 in the text.

We have now referred to Fig. 4 for those discussions.

Lines 304-307: It is not clear the canceling effect here. For the authors' arguments, the result should be seen in summer and winter.

To clarify our explanation, we have now rephrased the statements as "*Surprisingly, the modeled $N_C$ shows good agreement with observations, despite the overestimated $N_{CCN}$. One plausible reason is the canceling effect from the too strong $N_C$ sink in the model. The overestimated cloud droplet size by the model (Fig. 3b) fosters the warm rain formation, and in turn, efficiently deplete cloud droplets (Zheng et al., 2022b), keeping the modeled $N_C$ at a comparable level with the observations.*" In the summer, the modeled Nc is also comparable with observations (the red line stay in the shading of the observations), so the hypothesized canceling effect may take place as well.

Line 343: Since the authors defined CLWC in the paper, they should avoid using cloud LWC.

Revised as suggested.

Line 363: increased CLWC?

Corrected as suggested.

Line 398: SCAM5/6

Corrected as suggested.

Line 423: The link does not work.

The link is restored, and the related materials are available to view.

Figure 5: Why using normalized height for Nc?

The reason is that the cloud layer thickness and vertical positions differ for each corresponding time stamp. We need to normalize the height within each cloud layer to ensure that the Nc vertical variation is representative. We have now made it clear in the figure caption.